# Context Forcing: Consistent Autoregressive Video Generation with Long Context

Shuo Chen [* 1]   Cong Wei [* 2]   Sun Sun [2]   Tiancheng Shen [1]   Ping Nie [2]   Kai Zou [3]   Ge Zhang [4]   Ming-Hsuan Yang [1]
Wenhu Chen [2]

🌐 **Website:** https://chenshuo20.github.io/Context_Forcing

 **Code:** https://github.com/TIGER-AI-Lab/Context-Forcing

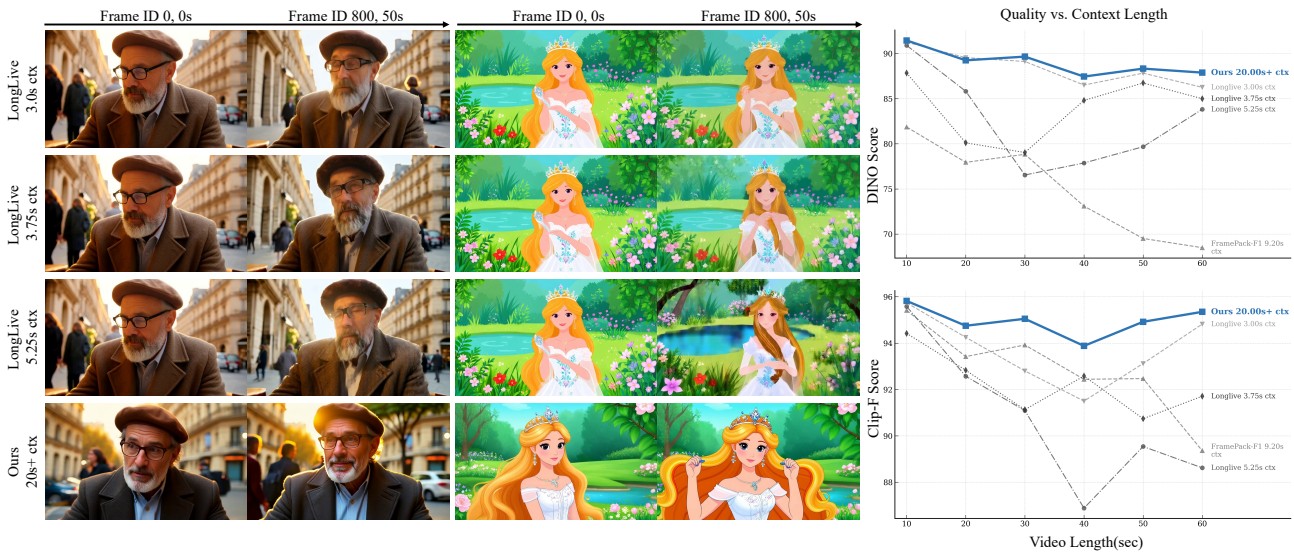

*Figure 1.* **Context Forcing mitigates the forgetting–drifting dilemma.** (1) State-of-the-art models are limited by short context windows (3.0–9.2 s), which leads to poor long-term consistency (*Forgetting*). (2) For streaming long-context tuning baselines (e.g., LongLive), enlarging the context window during inference (3.0 → 5.25 s) causes error accumulation and distribution shift (*Drifting*). In contrast, **Context Forcing** supports **20s+** context while maintaining strong long-term consistency.

## Abstract

Recent approaches to real-time long video generation typically employ streaming tuning strategies, attempting to train a long-context student using a short-context (memoryless) teacher. In these frameworks, the student performs long rollouts but receives supervision from a teacher limited to short 5-second windows. This structural discrepancy creates a critical **student-teacher mismatch**: the teacher's inability to access long-term history prevents it from guiding the student on global temporal dependencies, effectively capping the student's context length. To resolve this, we propose **Context Forcing**, a novel framework that trains a long-context student via a long-context teacher. By ensuring the teacher is aware of the full generation history, we eliminate the supervision mismatch, enabling the robust training of models capable of long-term consistency. To make this computationally feasible for extreme durations (e.g., 2 minutes), we introduce a context management system that transforms the lin-

---
[*]Equal contribution  [1]Department of EECS, University of California, Merced, USA  [2]University of Waterloo, Canada  [3]Netmind.AI  [4]M-A-P. Correspondence to: Ming-Hsuan Yang <mhyang@ucmerced.edu>, Wenhu Chen <wenhuchen@uwaterloo.ca>.

*Proceedings of the $43^{rd}$ International Conference on Machine Learning*, Seoul, South Korea. PMLR 306, 2026. Copyright 2026 by the author(s).

early growing context into a **Slow-Fast Memory** architecture, significantly reducing visual redundancy. Extensive results demonstrate that our method enables effective context lengths exceeding 20 seconds—$2$–$10\times$ longer than state-of-the-art methods like LongLive and Infinite-RoPE. By leveraging this extended context, Context Forcing preserves superior consistency across long durations, surpassing state-of-the-art baselines on various long video evaluation metrics.

## 1. Introduction

In recent years, video diffusion models based on architectures such as the Denoising Diffusion Transformer(DiT) (Peebles & Xie, 2023) have achieved remarkable success in generating photorealistic videos (Wan et al., 2025). While bidirectional models perform well for short clips, their computational cost limits long-form generation. To address this, the field is moving toward causal video architectures (Yin et al., 2024c; Huang et al., 2025), which, like Large Language Models, can theoretically generate infinite-length videos by predicting future frames from past context.

Despite this promise, current causal video models struggle to maintain coherence over long-term contexts. Effective context is often limited to just a few seconds (Cui et al., 2025; Yang et al., 2025; Zhang et al., 2026; Huang et al., 2025; Yesiltepe et al., 2025), beyond which identity shifts and temporal inconsistencies emerge. We identify the root cause as a fundamental **student-teacher mismatch**. As illustrated in Figure 2(b), current methods typically train a student to perform long rollouts using supervision from a memoryless teacher limited to short windows (e.g., 5 seconds). The teacher's inability to access long-term history prevents it from guiding the student on global temporal dependencies, effectively capping the student's learnable context length.

This mismatch results in a critical challenge for real-time long-context video generation, which we term the *Forgetting-Drifting Dilemma* (Figure 1). Existing methods face an unavoidable trade-off:

- **Forgetting:** Restricting the model to a short memory window minimizes error accumulation but causes the model to lose track of previous subjects and scenes during long rollout.
- **Drifting:** Maintaining a long context preserves identity but exposes the model to its own accumulated errors. Without a teacher capable of correcting these long-term deviations, the video distribution progressively drifts away from the real manifold.

To address these challenges, we propose **Context Forcing**, a framework that distills a long-context teacher into a long-context student. Our approach resolves the context-drifting dilemma by bridging the capability gap between teacher and student. We first leverage a Context Teacher pretrained on video continuation tasks, which is capable of processing long-context inputs. This teacher guides the student via *Contextual Distribution Matching Distillation*, explicitly transferring the ability to model long-term dependencies and ensuring global consistency. Furthermore, by exposing the student to imperfect, self-generated contexts during training, we enable it to actively recover from accumulated artifacts. The resulting robustness allows for $2-10\times$ longer duration Key-Value (KV) cache management (maintaining 20+ seconds of history) compared to prior SOTA (1.5–9.2 seconds of history) during inference, effectively addressing the forgetting-drifting trade-off and enabling consistent, long-form video generation.

The contributions of this work are:

- We introduce **Context Forcing**, a novel framework that mitigates the student-teacher mismatch in training real-time long video models. By distilling from a long-context teacher aware of the full generation history, we enable the robust training of a long-context student capable of long-term consistency.
- To support this, we design a context management system that transforms the linearly growing context into a **Slow-Fast Memory** architecture, significantly reducing visual redundancy. This mechanism enables effective context lengths exceeding 20 seconds—$2$–$10\times$ longer than state-of-the-art methods.
- We demonstrate that, equipped with these extended context lengths, our model preserves superior consistency across long durations, surpassing state-of-the-art baselines on various long video evaluation metrics.

## 2. Related Work

**Long Video Generation.** The high computational cost of Diffusion Transformers (DiTs) (Kong et al., 2024; Wan et al., 2025; Peebles & Xie, 2023; Yang et al., 2024) has limited video generation to short clips. To extend temporal horizons, many works combine diffusion with autoregressive (AR) prediction (Kim et al., 2024; Lin et al., 2025; Gu et al., 2025), including NOVA (Deng et al., 2024), Pyramid-Flow (Jin et al., 2024), and MAGI-1 (Teng et al., 2025). Other approaches improve efficiency via causal or windowed attention and KV caching (Yin et al., 2024c; Huang et al., 2025; Kodaira et al., 2025), or extend context through training-free positional encoding modifications (Lu et al., 2024; Lu & Yang, 2025; Zhao et al., 2025) or sparse attention routing module (Cai et al., 2025). However, most methods still struggle with global consistency beyond 10-

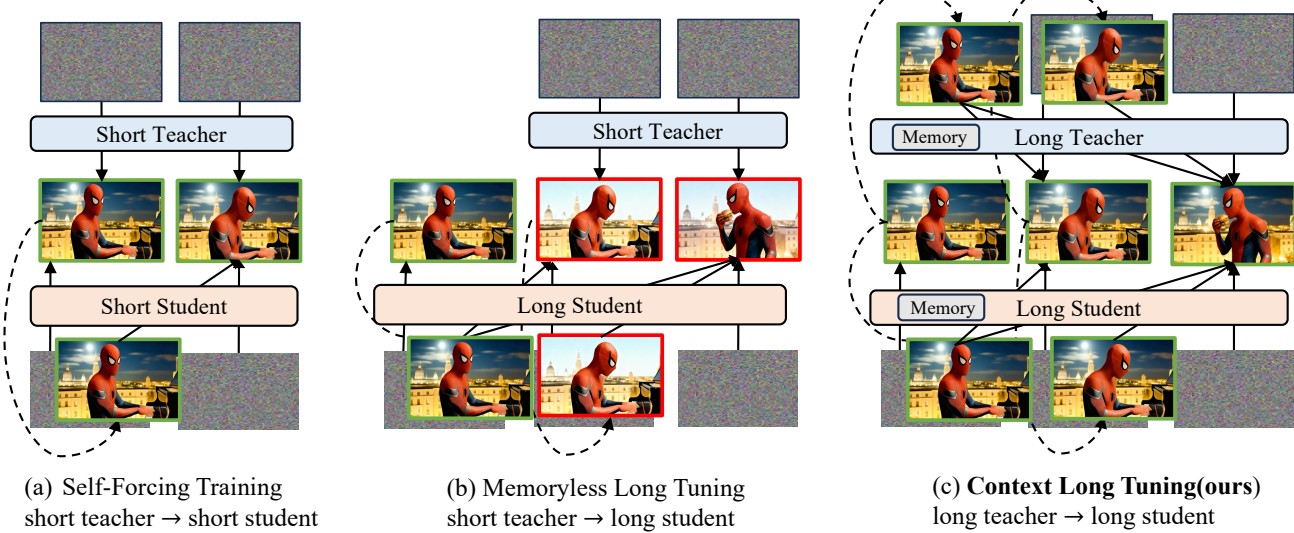

*Figure 2.* **Training paradigms for AR video diffusion models.** (a) Self-forcing: A student matches a teacher capable of generating only 5s video using a 5s self-rollout. (b) Longlive (Yang et al., 2025): The student performs long rollouts supervised by a memoryless 5s teacher on random chunks. The teacher's inability to see beyond its 5s window creates a student-teacher mismatch. (c) **Context Forcing (Ours)**: The student is supervised by a long-context teacher aware of the full generation history, resolving the mismatch in (b).

20 seconds. A key challenge of long video generation is error accumulation (drifting), addressed either during training by exposing models to drifted inputs (Cui et al., 2025; Chen et al., 2024; 2025) or during inference via recaching, sampling strategies, or feedback (Yang et al., 2025; Zhang et al., 2026; Li et al., 2025a; Ji et al., 2025). To enable real-time generation, recent works distill multi-step diffusion into few-step models (Valevski et al., 2024; Liu et al., 2023; Luo et al., 2023; Sauer et al., 2024), including Distribution Matching Distillation (DMD/DMD2) (Yin et al., 2024b;a; Wang et al., 2023) and Consistency Models (CM) (Song et al., 2023; Wang et al., 2024).

**Causal Video Generation.** Causal video generation synthesizes video sequences under strict temporal ordering constraints, thereby enabling streaming inference and long-horizon synthesis. Although early autoregressive models (Vondrick et al., 2016; Kalchbrenner et al., 2017) generated frames or tokens sequentially, they often suffered from error accumulation and poor scalability. Recent diffusion-based frameworks have improved visual fidelity by incorporating causal architectural priors, such as the block-wise causal attention introduced in CausVid (Yin et al., 2024c). To mitigate distribution shift, Self-Forcing (Huang et al., 2025), LongLive (Yang et al., 2025) and Self-Forcing++ (Cui et al., 2025) align training with inference by conditioning on prior outputs via KV caching and rollout-based objectives. InfinityRoPE (Yesiltepe et al., 2025) achieve a reduction of error accumulation by modifying positional encodings. Further research has addressed efficient long-context inference through windowed attention, as seen in StreamDiT (Kodaira et al., 2025).

**Memory Mechanism for Video Generation** Memory mechanisms are key to extending temporal context and maintaining consistency in long-horizon generation. ConsistI2V (Ren et al., 2024) uses explicit attention to first frame, Context as Memory (Yu et al., 2025), and World-Mem (Xiao et al., 2025) and Framepack (Zhang et al., 2026) introduce explicit memory structures to accumulate scene or contextual information over time, while RELIC (Hong et al., 2025) employs recurrent latent states for efficient long-range dependency modeling. PFP (Zhang et al., 2025) compress long videos into short context by training a novel compression module. Concurrent work WorldPlay (Sun et al., 2025) targets real-time interactive world modeling with long-term geometric consistency.

## 3. Methodology

We operate within the causal autoregressive framework, where the generation of a long video $X_{1:N}$ is decomposed into a sequence of conditional steps over frames or short chunks $X_t$. State-of-the-art methods, such as CausVid (Yin et al., 2024c) and Self-Forcing (Huang et al., 2025), enforce strict temporal causality via block-wise attention, modeling the distribution as $\prod_t p(X_t \mid X_{<t})$. These approaches typically employ Distribution Matching Distillation (DMD) (Yin et al., 2024b) to distill a high-quality bidirectional teacher into a causal student. Building on these foundations, we introduce **Context Forcing**.

Our goal is to train a causal video diffusion model, param-

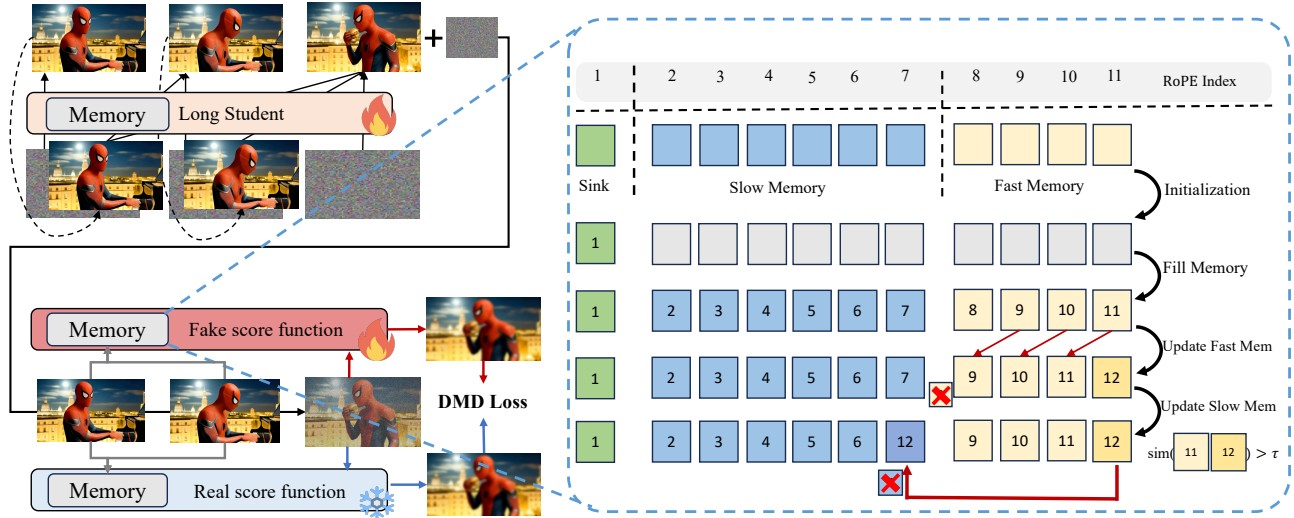

*Figure 3.* **Context Forcing and Context Management System.** We use KV Cache as the context memory, and we organize it into three parts: sink, slow memory and fast memory. During contextual DMD training, the long teacher provides supervision to the long student by utilizing the same context memory mechanism.

eterized by $\theta$, whose induced distribution over *long videos* $p_\theta(X_{1:N})$ matches the real data distribution $p_{\text{data}}(X_{1:N})$. Here, $N$ represents a duration spanning tens or hundreds of seconds. The objective is to minimize the global long-horizon KL divergence:

$$\mathcal{L}_{\text{global}} = \min_\theta \text{KL}\big(p_\theta(X_{1:N}) \,\|\, p_{\text{data}}(X_{1:N})\big). \quad (1)$$

Directly optimizing Eq. (1) ensures long-term coherence but is computationally intractable for large $N$. By applying the chain rule of KL divergence, we decompose the global objective into two components:

$$\mathcal{L}_{\text{global}} = \underbrace{\text{KL}\big(p_\theta(X_{1:k}) \,\|\, p_{\text{data}}(X_{1:k})\big)}_{\mathcal{L}_{\text{local}}: \text{ Local Dynamics}}$$
$$+ \underbrace{\mathbb{E}_{X_{1:k} \sim p_\theta}\Big[\text{KL}\big(p_\theta(X_{k+1:N}|X_{1:k}) \,\|\, p_{\text{data}}(X_{k+1:N}|X_{1:k})\big)\Big]}_{\mathcal{L}_{\text{context}}: \text{ Global Continuation Dynamics}}$$
$$(2)$$

This decomposition motivates our two-stage curriculum:

- **Stage 1 (Optimizing $\mathcal{L}_{\text{local}}$):** We match the distribution of short windows ($X_{1:k}$) to the real data distribution to learn local dynamics.
- **Stage 2 (Optimizing $\mathcal{L}_{\text{context}}$):** We match the model's continuation predictions ($X_{k+1:N}$) with the temporal evolution of real data to learn long-term dependencies.

### 3.1. Stage 1: Local Distribution Matching

The first stage warms up the causal student by minimizing $\mathcal{L}_{\text{local}}$. Given a teacher distribution $p_T(X_{1:k})$ (approximately the real data), we optimize:

$$\mathcal{L}_{\text{local}} = \text{KL}\big(p_\theta(X_{1:k}) \,\|\, p_T(X_{1:k})\big), \quad (3)$$

where $k$ corresponds to a 1–5 second window. We estimate the distribution matching gradient follow DMD (Yin et al., 2024b). Let $x = G_\theta(z)$ for noise $z$, and let $x_t$ be the diffused version of $x$ at timestep $t$. The gradient is given by:

$$\nabla_\theta \mathcal{L}_{\text{local}} \approx \mathbb{E}_{z,t,x_t}\Big[w_t \alpha_t \big(s_\theta(x_t, t) - s_T(x_t, t)\big) \frac{\partial G_\theta(z)}{\partial \theta}\Big], \quad (4)$$

where $s_\theta$ and $s_T$ are the student and teacher scores, respectively, and $w_t$ is a weighting function. This stage ensures $p_\theta(X_{1:k}) \approx p_{\text{data}}(X_{1:k})$, providing high-quality contexts for the subsequent stage.

### 3.2. Stage 2: Contextual Distribution Matching

Stage 2 targets $\mathcal{L}_{\text{context}}$, the second term of Eq. (2). This term requires minimizing the divergence between the student's continuation $p_\theta(\cdot|X_{1:k})$ and the true data continuation $p_{\text{data}}(\cdot|X_{1:k})$.

However, $p_{\text{data}}$ is not directly accessible for arbitrary contexts generated by the student. To solve this, we employ a pretrained **Context Teacher** $T$, which provides a reliable proxy distribution $p_T(X_{k+1:N} \mid X_{1:k})$. We rely on two key assumptions to justify using the teacher as a target:

**Assumption 1 (Teacher reliability near student contexts).** *Whenever the student context $X_{1:k} \sim p_\theta(X_{1:k})$ remains close to the real data manifold, the teacher's continuation $p_T(X_{k+1:N} \mid X_{1:k})$ is accurate.* This holds whenever the teacher is well-trained on real video prefixes.

**Assumption 2 (Approximate real prefixes).** *Stage 1 successfully aligns $p_\theta(X_{1:k})$ with $p_{data}(X_{1:k})$.* This ensures that student rollouts remain within the teacher's reliable region during Stage 2 training.

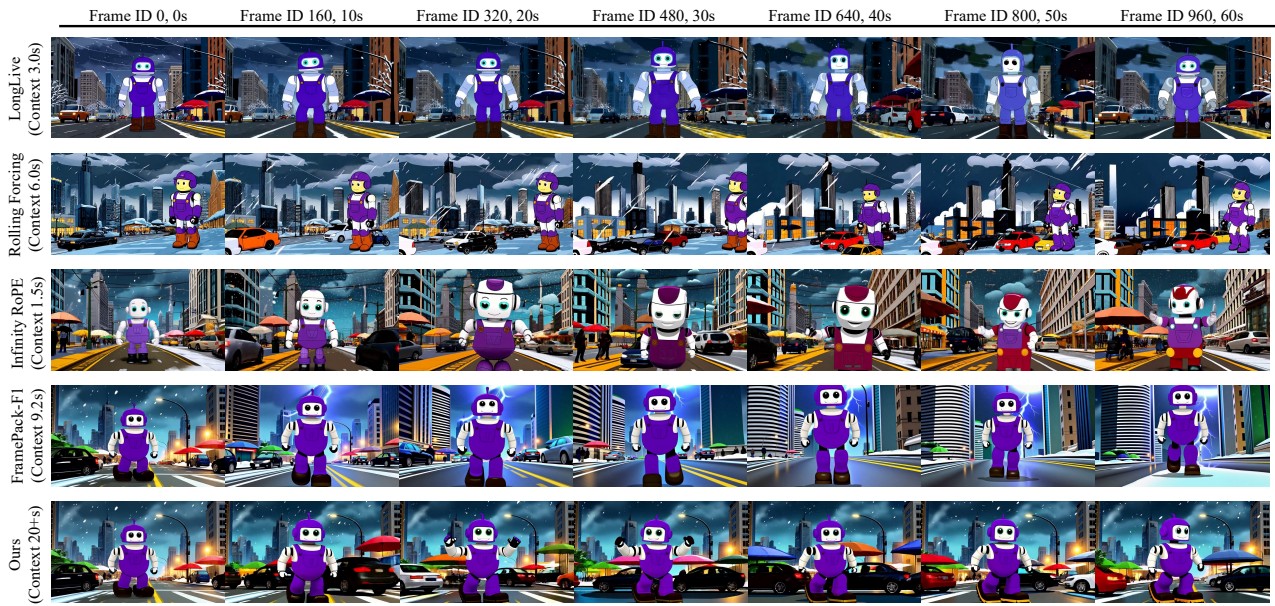

*Figure 4.* **Comparison on 1-min Video Generation.** Our method keeps both the background and subject consistent across 1-min video, while other baselines have different levels drifting or identity shift.

Under these assumptions, we approximate $p_{\text{data}} \approx p_T$ in the second term of Eq. (2), yielding the **Contextual DMD (CDMD)** objective:

$$
\mathcal{L}_{\text{CDMD}} = \mathbb{E}_{X_{1:k} \sim p_\theta(X_{1:k})}
$$
$$
\Big[ \text{KL}\big(p_\theta(X_{k+1:N} \mid X_{1:k}) \,\|\, p_T(X_{k+1:N} \mid X_{1:k})\big) \Big]
$$
(5)

Crucially, the expectation is over $X_{1:k} \sim p_\theta$, ensuring the student is trained on its *own* rollouts, thereby mitigating exposure bias.

**Score-based CDMD Gradient.** We estimate the gradient of Eq. (5) using a conditional variant of the DMD gradient. Let $x_{\text{cont}} = G_\theta(z_{\text{cont}} \mid X_{1:k})$ be the generated continuation, and $x_{t,\text{cont}}$ be its diffused version. Running both fake score and real score models on the *same* student-generated context produces scores $s_\theta(\cdot \mid X_{1:k})$ and $s_T(\cdot \mid X_{1:k})$. The gradient is:

$$
\nabla_\theta \mathcal{L}_{\text{CDMD}} \approx \mathbb{E}_{\substack{X_{1:k} \sim p_\theta \\ z_{\text{cont}}, t}} \Big[ w_t \alpha_t \big( s_\theta(x_{t,\text{cont}}, t \mid X_{1:k})
$$
$$
- s_T(x_{t,\text{cont}}, t \mid X_{1:k}) \big) \frac{\partial G_\theta(z_{\text{cont}} \mid X_{1:k})}{\partial \theta} \Big].
$$
(6)

By descending Eq. (6), we align the student's long-term autoregressive dynamics with the teacher's robust priors.

**Long Self-Rollout Curriculum.** Minimizing $\mathcal{L}_{\text{context}}$ requires the context horizon $k$ to approach the full sequence length $N$. However, sampling $X_{1:k} \sim p_\theta$ for large $k$ early in training causes severe distribution shift due to accumulated drift. To mitigate this, we employ a dynamic horizon sched-

ule $N_{\max}^{(t)}$ that grows linearly with training step $t$. At each iteration, the rollout length is sampled as $k \sim \mathcal{U}(k_{\min}, N_{\max}^{(t)})$. This curriculum initializes training in the stable Stage 1 regime ($k \approx k_{\min}$) and progressively exposes the model to long-range dependencies.

**Clean Context Policy.** Self Forcing (Huang et al., 2025) typically generates rollouts using a random timestep selection strategy to ensure supervision across all diffusion steps. We retain this random exit policy for the *target* frames $X_{k+1:N}$ to preserve gradient coverage, but enforce that the *context* frames $X_{1:k}$ are fully denoised. We apply a complete few-step denoising process to the context. This decoupling ensures the context remains informative and aligned with the teacher's training distribution but also maintains supervision for every diffusion step.

### 3.3. Context Management System

Our teacher and student models share an identical architecture; both are autoregressive generative models augmented with a memory module for context retention. We utilize KV caches to represent the context $X_{1:k}$. To maintain efficiency as the sequence length $k$ grows, we design a KV cache management system inspired by dual-process memory theories. Specifically, the cache $\mathcal{M}$ is partitioned into three functional components: an *Attention Sink*, *Slow Memory* (Context), and *Fast Memory* (Local). Both the student and teacher are equipped with this system.

**Cache Partitioning.** The total cache is defined as the union

Frame ID 0, 0s    Frame ID 160, 10s    Frame ID 320, 20s    Frame ID 480, 30s    Frame ID 640, 40s    Frame ID 800, 50s    Frame ID 960, 60s

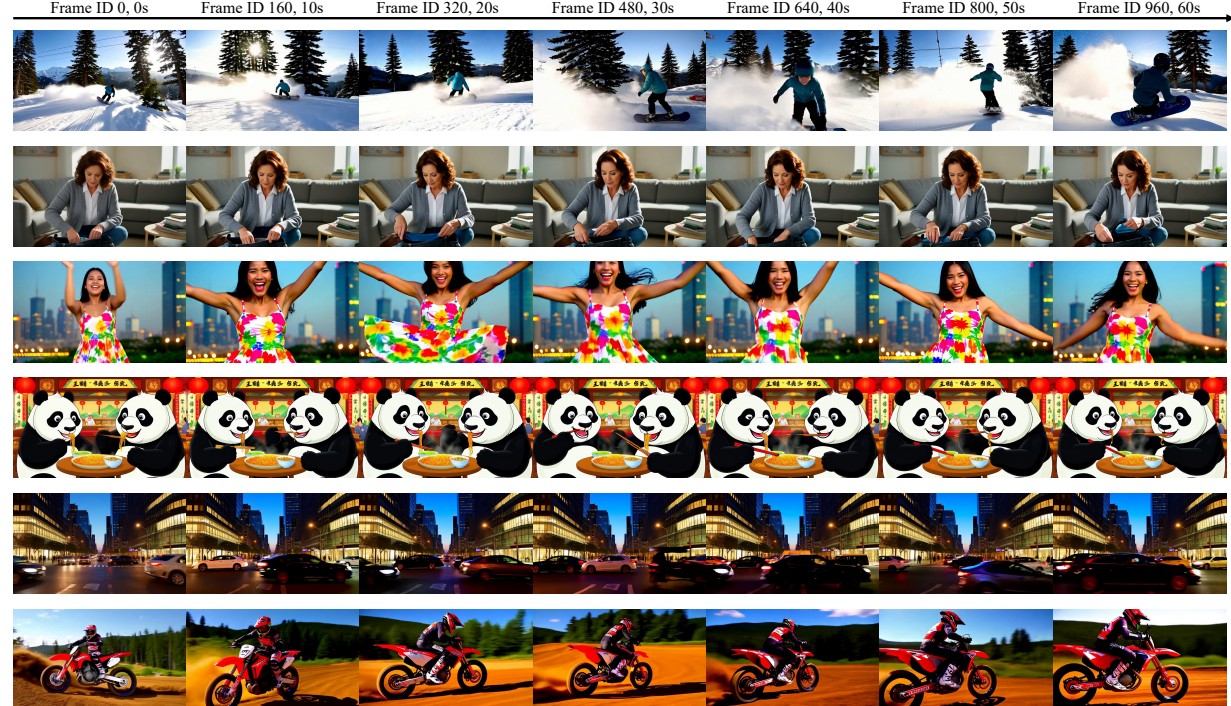

*Figure 5.* **Qualitative Results of Context Forcing.** Our method enables minute-level video generation with minimal drifting and high consistency across diverse scenarios.

of disjoint sets:

$$\mathcal{M} = \mathcal{S} \cup \mathcal{C}_{\text{slow}} \cup \mathcal{L}_{\text{fast}}.$$

- *Attention Sink ($\mathcal{S}$)*: Retains initial $N_s$ tokens to stabilize attention, following StreamingLLM (Yang et al., 2025; Shin et al., 2025).
- *Slow Memory ($\mathcal{C}_{slow}$)*: A long-term buffer of up to $N_c$ tokens, storing high-entropy keyframes and updating only with significant new information.
- *Fast Memory ($\mathcal{L}_{fast}$)*: A rolling FIFO queue of size $N_l$, capturing immediate local context with short-term persistence.

**Surprisal-Based Consolidation.** Upon generating a new token $x_t$ and enqueuing it into the Fast Memory $\mathcal{L}_{\text{fast}}$, we evaluate its informational value relative to the immediate temporal context. We postulate that tokens exhibiting high similarity to their predecessors carry redundant information (low surprisal), whereas dissimilar tokens indicate significant state transitions or visual changes (high surprisal).

To capture these high-information moments efficiently, we compare the key vector $k_t$ of the current token with that of the immediately preceding token $k_{t-1}$. The consolidation policy $\pi(x_t)$ determines whether $x_t$ is promoted to Slow Memory $\mathcal{C}_{\text{slow}}$:

$$\pi(x_t) = \begin{cases} \text{Consolidate} & \text{if } \text{sim}(k_t, k_{t-1}) < \tau, \\ \text{Discard} & \text{otherwise,} \end{cases} \quad (7)$$

where $\tau$ is a similarity threshold. This criterion ensures that $\mathcal{C}_{\text{slow}}$ prioritizes storing temporal gradients and distinctive events rather than static redundancies. As with standard cache management, if $|\mathcal{C}_{\text{slow}}| > N_c$ after consolidation, the oldest entry is evicted to maintain fixed memory complexity.

**Bounded Positional Encoding.** Unlike standard autoregressive video models (Huang et al., 2025; Cui et al., 2025), where positional indices grow unbounded ($p_t = t \rightarrow \infty$), leading to distribution shifts on long sequences, we adopt *Bounded Positional Indexing*. All tokens' temporal RoPE positions are constrained to a fixed range $\Phi = [0, N_s + N_c + N_l - 1]$ regardless of generation step $t$:

$$\phi(x) = \begin{cases} i \in [0, N_s - 1] & \text{if } x \in \mathcal{S}, \\ j \in [N_s, N_c - 1] & \text{if } x \in \mathcal{C}_{\text{slow}}, \\ k \in [N_c, N_c + N_l - 1] & \text{if } x \in \mathcal{L}_{\text{fast}}. \end{cases} \quad (8)$$

This creates a static attention window where recent history (Fast) slides through high indices, while salient history (Slow) is compressed into lower indices, stabilizing attention over long sequences.

### 3.4. Robust Context Teacher Training

Standard training conditions the model on ground-truth context, but inference relies on self-generated history, creating a distribution shift known as exposure bias. To ensure our Context Teacher provides robust guidance even when

Calculate error: $e_{drift} = X_{Pred} - X_{GT}$

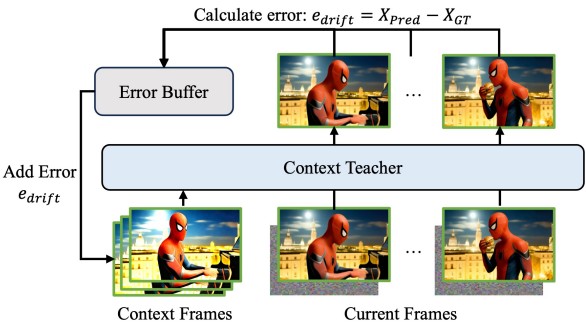

*Figure 6.* **Robust Context Teacher Training via Error-Recycling Fine-Tuning.** To combat exposure bias, past predictive errors are dynamically stored in an Error Buffer and probabilistically injected into clean context and noise latents. The input sequence aligns directly with the student model's context management system.

the student drifts, we adopt Error-Recycling Fine-Tuning (ERFT) (Li et al., 2025a).

Rather than training on clean history $X_{1:k}$, we inject realistic accumulated errors into the teacher's context. We construct a perturbed context $\tilde{X}_{1:k} = X_{1:k} + \mathbb{I} \cdot e_{\text{drift}}$, where $e_{\text{drift}}$ is sampled from a bank of past model residuals and $\mathbb{I}$ is a Bernoulli indicator. The teacher is optimized to recover the correct velocity $v_{\text{target}}$ from $\tilde{X}_{1:k}$. This active correction capability ensures $p_T(\cdot \mid X_{1:k})$ remains a reliable proxy for $p_{\text{data}}$ even when the student's context $X_{1:k}$ degrades.

## 4. Experiments

**Implementation Details.** We implement the robust context teacher using Wan2.1-T2V-1.3B (Wan et al., 2025) as the base model. To construct the training dataset, we filter the Sekai (Li et al., 2025b) and Ultravideo (Xue et al., 2025) collections to retain high-quality videos exceeding 10 seconds in duration, yielding a total of 40k clips. The robust context teacher is trained for 8k steps with a batch size of 8. During training, frames are sampled uniformly from the 5–20 second interval of the video data to serve as context.

The student model also utilizes the Wan2.1-T2V-1.3B model. In Stage 1, we employ 81-frame video clips from the Vid-ProM (Wang & Yang, 2024) dataset and train for 600 iterations with a batch size of 64. In Stage 2, which focuses on context distillation, we extend the rollout horizon to video lengths of 10–30 seconds to address short-term memory limitations. This phase is similarly trained on the VidProM dataset for 500 iterations using the same batch size. For both teacher and student models, we set the KV cache size to 21 latent frames, and set $N_s = 3, N_c = 12, N_l = 6, \tau = 0.95$. We implement Surprisal-Based Consolidation at 2-chunk intervals. Upon chunk consolidation, we retain only the first latent, effectively extending the context beyond 20s.

**Baselines.** We evaluate our method against three distinct categories of baselines. The first category comprises bidi-

Student Generate Frames        Context Teacher Generated Frames

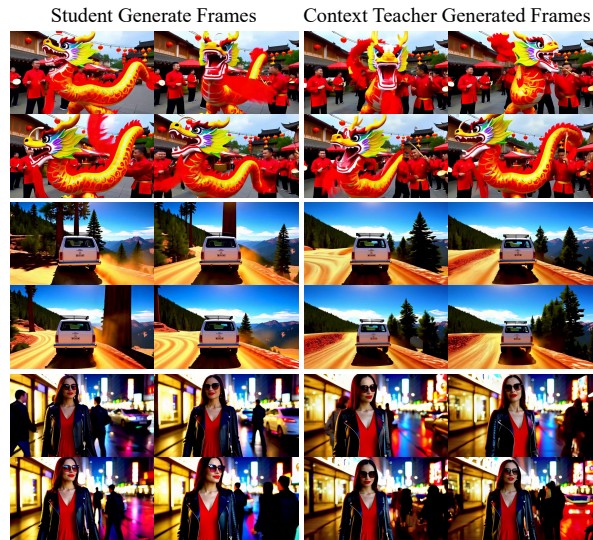

*Figure 7.* **Video Continuation with Robust Context Teacher.** Context teacher can generate next segment videos with context generated by student.

rectional diffusion models, specifically LTX-Video (Ha-Cohen et al., 2024) and Wan2.1 (Wan et al., 2025). The second category includes autoregressive models such as SkyReels-V2 (Chen et al., 2025), MAGI-1 (Teng et al., 2025), CausVid (Yin et al., 2024c), NOVA (Deng et al., 2024), Pyramid-Flow (Jin et al., 2024), and Self Forcing (Huang et al., 2025). The third category consists of recent methods targeting long video generation within autoregressive frameworks. These include LongLive (Yang et al., 2025) with a context length of 3 seconds, Self Forcing++ (Cui et al., 2025), Rolling Forcing (Liu et al., 2025) with a context length of 6 seconds, and Infinity-RoPE (Yesiltepe et al., 2025) with a context length of 1.5 seconds. Finally we include a long context baseline Framepack (Zhang et al., 2026) with a context length of 9.2 seconds.

**Evaluation.** We report performance on VBench (Zheng et al., 2025) following (Huang et al., 2025; Yang et al., 2025). Beyond standard benchmarks, we assess fine-grained consistency using DINOv2 (Oquab et al., 2023) (structural identity), CLIP-F (Radford et al., 2021) (semantic context), and CLIP-T (prompt alignment). To improve robustness against temporal artifacts, we implement window-based sampling: for any timestamp $t$, we compute the average cosine similarity between the first frame ($V_0$) and frames within $[t - 0.5s, t + 0.5s]$. We average results over five random seeds per prompt to ensure statistical reliability. This approach effectively measures long-term subject and background consistency.

### 4.1. Video Continuation with Robust Context Teacher

To evaluate the context teacher, we feed the teacher model with videos generated by the student model after Stage 1

*Table 1.* Single-prompt 60-second long video consistency evaluation.

| Model | Context Length ↑ | Dino Score ↑ | | | | | | Clip-F Score ↑ | | | | | | Clip-T Score ↑ | | | | | | Background Consistency↑ | Subject Consistency↑ |
|---|---|---|---|---|---|---|---|---|---|---|---|---|---|---|---|---|---|---|---|---|---|
| | | 10s | 20s | 30s | 40s | 50s | 60s | 10s | 20s | 30s | 40s | 50s | 60s | 10s | 20s | 30s | 40s | 50s | 60s | | |
| FramePack-F1 | 9.2s | 81.86 | 77.95 | 78.84 | 73.10 | 69.52 | 68.50 | 95.41 | 93.42 | 93.92 | 92.44 | 92.47 | 89.36 | 36.36 | 34.67 | 33.75 | 33.84 | 34.77 | 32.30 | 91.61 | 89.15 |
| LongLive | 3.0s | 91.25 | 89.55 | 89.12 | 86.51 | 87.83 | 86.26 | 95.74 | 94.25 | 92.80 | 91.50 | 93.12 | 94.82 | 36.95 | 35.80 | 36.17 | 36.58 | 35.92 | 37.13 | 94.92 | 93.05 |
| Infinate RoPE | 1.5s | 91.18 | 88.17 | 85.37 | 79.80 | 81.10 | 83.72 | 94.09 | 91.71 | 91.13 | 89.71 | 86.11 | 88.88 | 35.26 | 35.03 | 35.88 | 32.56 | 32.29 | 32.28 | 92.42 | 90.11 |
| **Ours, teacher** | 20+s | 87.61 | - | - | - | - | - | 95.52 | - | - | - | - | - | 35.93 | - | - | - | - | - | 95.24 | 94.87 |
| **Ours, student** | 20+s | 91.45 | 89.25 | 89.66 | 87.45 | 88.33 | 87.89 | 95.82 | 94.75 | 95.05 | 93.88 | 94.92 | 95.35 | 37.12 | 36.25 | 36.72 | 37.15 | 37.08 | 37.66 | 95.95 | 95.68 |

*Table 2.* Comparison of video generation models across architecture families.

| Model | #Params | Throughput (FPS) ↑ | Evaluation scores on 5s ↑ | | | | | Evaluation scores on 60s ↑ | | | | |
|---|---|---|---|---|---|---|---|---|---|---|---|---|
| | | | Total | Quality | Semantic | Background Consistency | Subject Consistency | Total | Quality | Semantic | Background Consistency | Subject Consistency |
| *Bidirectional models* | | | | | | | | | | | | |
| LTX-Video | 1.9B | 8.98 | 80.00 | 82.30 | 70.79 | 95.30 | 95.01 | - | - | - | - | - |
| Wan2.1 | 1.3B | 0.78 | 84.26 | 85.30 | 80.09 | 96.96 | 95.99 | - | - | - | - | - |
| *Autoregressive models* | | | | | | | | | | | | |
| SkyReels-V2 | 1.3B | 0.49 | 82.67 | 84.70 | 74.53 | 96.83 | 96.07 | 70.47 | 75.30 | 51.15 | 89.95 | 84.99 |
| MAGI-1 | 4.5B | 0.19 | 79.18 | 82.04 | 67.74 | 96.83 | 95.83 | 69.87 | 76.12 | 44.87 | 87.76 | 79.46 |
| CausVid | 1.3B | 17.0 | 81.20 | 84.05 | 69.80 | 95.12 | 95.96 | 71.04 | 76.80 | 48.01 | 89.85 | 86.75 |
| NOVA | 0.6B | 0.88 | 80.12 | 80.39 | 79.05 | 95.16 | 93.38 | 65.25 | 70.25 | 45.24 | 88.06 | 77.50 |
| Pyramid Flow | 2B | 6.7 | 81.72 | 84.74 | 69.62 | 96.09 | 96.08 | - | - | - | - | - |
| Self Forcing, chunk-wise | 1.3B | 17.0 | 84.31 | 85.07 | 81.28 | 95.98 | 96.29 | 71.86 | 77.20 | 50.51 | 87.84 | 83.60 |
| *Long autoregressive models* | | | | | | | | | | | | |
| LongLive | 1.3B | 20.7 | 84.87 | 86.97 | 76.47 | 96.55 | 95.82 | 83.64 | 84.53 | 74.97 | 94.62 | 93.88 |
| Self Forcing++ | 1.3B | 17.0 | 83.11 | 83.79 | 80.37 | - | - | - | - | - | - | - |
| Rolling Forcing | 1.3B | 15.8 | 81.22 | 84.08 | 69.78 | 96.11 | 96.02 | 79.31 | 81.87 | 67.69 | 94.12 | 93.10 |
| Infinity-RoPE | 1.3B | 17.0 | 81.79 | 83.27 | 75.87 | 96.34 | 95.14 | 79.99 | 80.81 | 74.30 | 94.21 | 93.05 |
| **Ours, student model** | 1.3B | 17.0 | 83.44 | 84.98 | 77.29 | 97.38 | 96.84 | 82.45 | 83.55 | 76.10 | 95.34 | 94.88 |

training. We then assess the consistency of the complete sequence, which comprises the initial context with the generated continuation. Evaluation is performed using 100 text prompts randomly sampled from MovieGenBench (Polyak et al., 2024). As illustrated in Figure 7, the context teacher effectively synthesizes the subsequent video segment, providing empirical support for Assumptions 1 and 2. Furthermore, we quantitatively evaluate the performance of the context teacher using student-generated videos as input, reporting subject and background consistency on VBench, as well as DINOv2, CLIP-F, and CLIP-T scores. The consistency metrics for the complete 10-second sequence are presented in Table 1, further demonstrating that the context teacher consistently produces reliable continuations from student-generated contexts.

### 4.2. Text-to-Short Video Generation

**Quantitative Results.** We quantitatively compare our method against baselines. We evaluate 5-second video generation on the VBench dataset using its official extended prompts. The results summarized in Table 2 demonstrate that our method achieves performance comparable to the baselines on short video generation.

### 4.3. Text-to-Long Video Generation

**Qualitative Results.** We evaluate our proposed method against baseline models on 60-second video generation, with qualitative results illustrated in Figure 2. By leveraging a slow-fast memory architecture with a KV cache size of 21

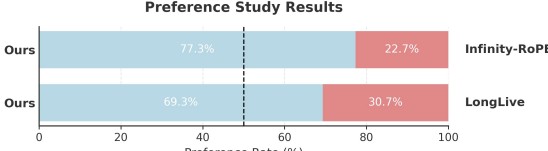

*Figure 8.* **Preference Study Results:** Our method significantly outperforms baseline models, achieving a 77.3% preference rate against Infinity-RoPE and a 69.3% preference rate against LongLive.

and a context span exceeding 20s, our method achieves superior consistency and effectively mitigates content drifting compared to the baselines.

**Quantitative Results.** We evaluate 60-second video generation performance on the VBench with results summarized in Table 2, using its offical extened prompts. Additionally, we report DINOv2, CLIP-F, and CLIP-T scores in Table 1, using 100 text prompts randomly sampled from MovieGen-Bench (Polyak et al., 2024), following the same experimental protocol as in Section 4.1. Both tables demonstrate that our method achieves high consistency, particularly during extended video sequences. Notably, while LongLive also achieves competitive scores, qualitative inspection reveals that it frequently exhibits abrupt scene resets and cyclic motion patterns, shown in Figure 10 in Appendix.

### 4.4. User Study

We designed and conducted a blind A/B testing user study via a web interface with 15 participants. We randomly se-

*Table 3.* Ablation study on Slow Memory Sampling Strategy, Context DMD, and Bounded Positional Encoding (evaluated on 60s).

| Model | Total Score ↑ | Quality Score ↑ | Semantic Score ↑ | Background Consistency ↑ | Subject Consistency ↑ | Dynamic Degree ↑ |
|---|---|---|---|---|---|---|
| *Slow Memory Sampling Strategy* | | | | | | |
| Uniform sample, interval 1 | 80.82 | 82.20 | 75.32 | 92.45 | 92.10 | 52.15 |
| Uniform sample, interval 2 | 81.11 | 82.61 | 75.12 | 93.12 | 92.85 | 55.30 |
| *Contextual Distillation* | | | | | | |
| w/o. Contextual Distillation | 80.36 | 82.28 | 72.70 | 93.55 | 93.20 | 48.12 |
| *Bounded Positional Encoding* | | | | | | |
| w/o. Bounded Positional Encoding | 73.52 | 75.44 | 65.82 | 84.68 | 79.24 | 27.45 |
| **Ours** | **82.45** | **83.55** | **76.10** | **95.34** | **94.88** | **58.26** |

Context Teacher Generate Frames(0-30s)

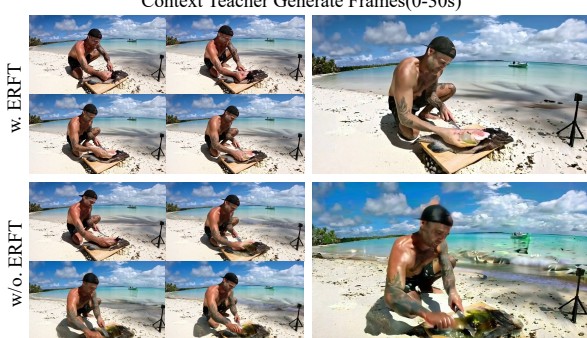

*Figure 9.* **Ablation on Error-Recycling Fine-Tuning (ERFT).** With ERFT, context teacher is more robust to accumulate error.

lected 20 text prompts and generated corresponding long videos using LongLive, Infinity-RoPE, and our Context Forcing method. Participants were shown paired videos (Ours vs. Baseline) in a randomized order. To guide their evaluation, participants were asked to comprehensively consider (1) overall visual quality, (2) long-term temporal consistency, and (3) semantic alignment, and then select their single most preferred result.

The overall preference (win rate) is summarized in the Figure 8, demonstrating a clear user preference for our method due to its stable long-term generation.

### 4.5. Ablation Studies

**Slow Memory Sampling Strategy** Our method employs a selection strategy based on key-vector similarity to sample context from slow memory. Unlike fixed uniform sampling, this strategy dynamically selects historical chunks that exhibit low similarity to the current generation window, thereby preserving critical semantic information over time. We compare our approach against alternative baselines, specifically uniform sampling with intervals of 1 and 2 chunks. As summarized in Table 3, the results demonstrate the effectiveness of similarity-based selection in maintaining long-term consistency.

**Context DMD Distillation** We evaluate the contribution of Contextual Distribution Matching Distillation by comparing our full model against a training-free baseline. In

the latter, our context management system is applied directly after Stage 1 training without the DMD process. The results in Table 3 indicate that removing Context DMD leads to a degradation in both semantic and temporal consistency, highlighting its critical role in enabling coherent, long-horizon video generation.

**Error-Recycling Fine-Tuning (ERFT).** We test the context teacher by taking 5s videos from the video dataset as input for autoregressive rollout. As shown in Figure 9, the visualization of 30s generation results indicates that with robust context training, the context teacher produces videos with fewer artifacts. This results in a better distribution for further contextual distillation.

**Bounded Positional Encoding.** We investigate the impact of Bounded Positional Encoding by excluding it during inference, with quantitative results presented in Table 3. In the absence of this encoding, we observe a significant performance drop in both background stability and subject consistency. This demonstrates its essential role in stabilizing long-range attention and mitigating temporal drift during the generation process.

## 5. Conclusion

In this work, we introduced **Context Forcing**, a framework designed to overcome the fundamental **student-teacher mismatch** in long-horizon causal video generation. By ensuring the teacher model maintains awareness of long-term history, our approach eliminates the supervision gap that limits existing streaming-tuning methods. To handle the computational demands of extreme durations, we proposed a **Slow-Fast Memory** architecture that effectively reduces visual redundancy. Extensive experiments demonstrate that Context Forcing achieves effective context lengths of 20+ seconds, a 2–10× improvement over current state-of-the-art baselines.

## 6. Limitations and Future Work

While context forcing demonstrates improved consistency and substantially reduced drifting in long-context scenarios, it is not entirely immune to errors. Residual drifting issues can still persist in particularly complex cases, and the overall fidelity is not absolute, with fine-grained details and subtle textures occasionally being omitted or distorted during generation. We believe these limitations stem in part from the inherent difficulty of compressing long-range temporal information without incurring any loss. Future research can investigate more advanced context compression techniques to enhance long-term information retention and further improve robustness against error drifting, ultimately enabling more faithful and stable generation over extended sequences.

## Impact Statement

This paper contributes to the advancement of generative AI by enhancing temporal consistency in long video generation. Our work enables the creation of more coherent and realistic visual sequences, which has significant positive potential in digital storytelling, filmmaking, world model and professional video editing. However, we acknowledge that the ability to generate highly consistent long-form videos also increases the risk of creating sophisticated synthetic media or deepfakes that could be used for misinformation. To mitigate these ethical concerns, we advocate for the integration of digital watermarking and provenance standards in downstream applications. We believe that fostering transparency and developing robust detection mechanisms are essential as video generation technology continues to mature.

## Acknowledgments

This work was supported in part by the Institute of Information & Communications Technology Planning & Evaluation (IITP) grant funded by the Korean Government (MSIT) (No. RS-2024-00457882, National AI Research Lab Project). This research was also supported by the New Beginnings program of the National Research Council of Canada (INBR6-001156).

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

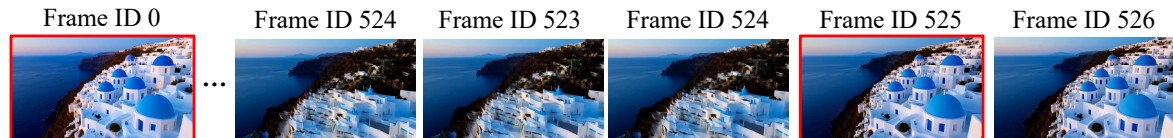

*Figure 10.* **Visual artifacts in LongLive.** The model exhibits a sudden flashback artifact, where the video abruptly resets to the initial frame after 524 frames, disrupting temporal continuity.

## A. Preliminaries

**Causal Autoregressive Models.** Causal autoregressive models generate videos at the frame or short-chunk level ($X_t$) while enforcing strict temporal causality. Methods such as CausVid (Yin et al., 2024c) and Self-Forcing (Huang et al., 2025) adopt block-wise causal attention, allowing bidirectional self-attention within each chunk $X_t$ but restricting information flow across chunks. Video generation is formulated as $P(X_t \mid X_{<t})$. In Self-Forcing, the student model is stochastically conditioned on its own generated outputs $\hat{X}_{<t}$ during training. These models typically employ Distribution Matching Distillation (DMD) (Yin et al., 2024b) to distill knowledge from a bidirectional teacher into a causal student.

## B. Visual artifacts in LongLive.

While LongLive achieves respectable quantitative scores, we observe that it frequently suffers from abrupt scene resets and repetitive, cyclic motion patterns, as illustrated in Figure 10.

## C. Teacher Training Details

For each ground-truth video longer than 10 seconds, we prefill a 21-frame context memory system (3 sink, 12 slow-memory, and 6 fast-memory), aligning with the student's memory system. The subsequent 21 latent frames in the video then serve as the prediction target. To better approximate under exposure bias, we apply Error-Recycling Fine-Tuning, where past model residuals are defined as the teacher's own latent prediction errors rather than hand-crafted perturbations. Concretely, at timestep, we obtain the predicted latent via one-step ODE integration and define, where is the clean target latent. These residual tensors are computed online, stored in a timestep-indexed shared buffer, and uniformly resampled and probabilistically injected into the context memory during teacher training. This exposes the teacher to its own historical drift and trains it to recover from it. Please refer to Fig. 6 on the website for the pipeline.

## D. Consistency Evaluation by VLM

Following (Cai et al., 2026), we use Gemini-3-Pro to evaluate video consistency on 100 text prompts across long (30s) and short (5s) video settings. As shown in Table 4, our Teacher model achieves a score of 64.20 in the challenging 30-second setting, outperforming existing baselines like LongLive and Infinity-RoPE, while our distilled Student model maintains highly competitive performance. Furthermore, in the standard 5-second setting, our approach surpasses the Wan2.1-T2V-1.3B baseline, demonstrating robust generalization across both temporal horizons.

*Table 4.* Comparison of VLM consistency in 30s and 5s settings.

| Method (30s setting) | VLM consistency ↑ |
| --- | --- |
| Self Forcing(1.3B) (14B teacher) | 30.39 |
| LongLive(1.3B) (14B teacher) | 54.22 |
| Infinity-RoPE(1.3B) (14B teacher) | 53.83 |
| **Ours Teacher(1.3B)** | **64.20** |
| **Ours Student(1.3B) (1.3B teacher)** | 62.53 |

| Method (5s setting) | VLM consistency ↑ |
| --- | --- |
| Wan2.1-T2V-1.3B | 66.54 |
| **Ours Teacher(1.3B)** | **67.53** |

# E. Algorithm of Context Forcing.

Algorithm block of context forcing training.

---

**Algorithm 1** Contextual DMD

---

**Require:** Denoise timesteps $\{t_1, .., t_T\}$
**Require:** Pre-trained teacher $s_{real}$
**Require:** Checkpoints from stage 1, student score function $s_{fake}$, AR diffusion model $G_\phi$
**Require:** Text prompt dataset $\mathcal{D}$, rollout decay step $s_d$, rollout range $(L_0, L_1)$, context window $c$, teacher length $l$, local attention size $a$
  1: **Initialize**, step $s = 0$
  2: **Initialize** model output $X \leftarrow []$
  3: **Initialize** KV cache $C \leftarrow []$
  4: **while** training **do**
  5:     **Sample** prompt $p \sim \mathcal{D}$
  6:     **Sample** rollout length $L = \text{Uniform}(L_0, \frac{s}{s_d} \times (L_1 - L_0) + L_0 + 1)$
  7:     **Sample** random exit $r = \text{Uniform}(1, 2, ..., T)$
  8:     **for** $i = 1, ..., L$ **do**
  9:         **Initialize** $x_t^i \sim \mathcal{N}(0, \text{I})$
10:         **if** $L - r - l \leq i < L - l$ **then**
11:             $r' = T$
12:         **else**
13:             $r' = r$
14:         **end if**
15:         **for** $j = 1, ..., r'$ **do**
16:             **if** $j = r'$ **then**
17:                 Enable gradient computation
18:                 $\hat{x}_0^i \leftarrow G_\phi(x_{t_j}^i, t_j, C)$
19:                 $X.\text{append}(\hat{x}_0^i)$
20:                 Disable gradient computation
21:                 $C \leftarrow G_\phi^C(\hat{x}_0^i, 0, C)$
22:             **else**
23:                 Disable gradient computation
24:                 $\hat{x}_0^i = G_\phi(x_{t_j}^i, t_j, C)$
25:                 **Sample** $\epsilon \sim \mathcal{N}(0, \text{I})$
26:                 Set $x_{t_{j-1}}^i \leftarrow \text{addnoise}(\hat{x}_0^i, \epsilon, t_{j-1})$
27:             **end if**
28:         **end for**
29:         context video $v_c = X[L - r - l : L - l]$, target noise $v_t = \text{addnoise}(X[L - l :], t)$
30:         Compute Contextual DMD Loss with $s_{fake}(v_t, t, v_c)$ and $s_{real}(v_t, t, v_c)$
31:     **end for**
32: **end while**

---

# F. User Study UI

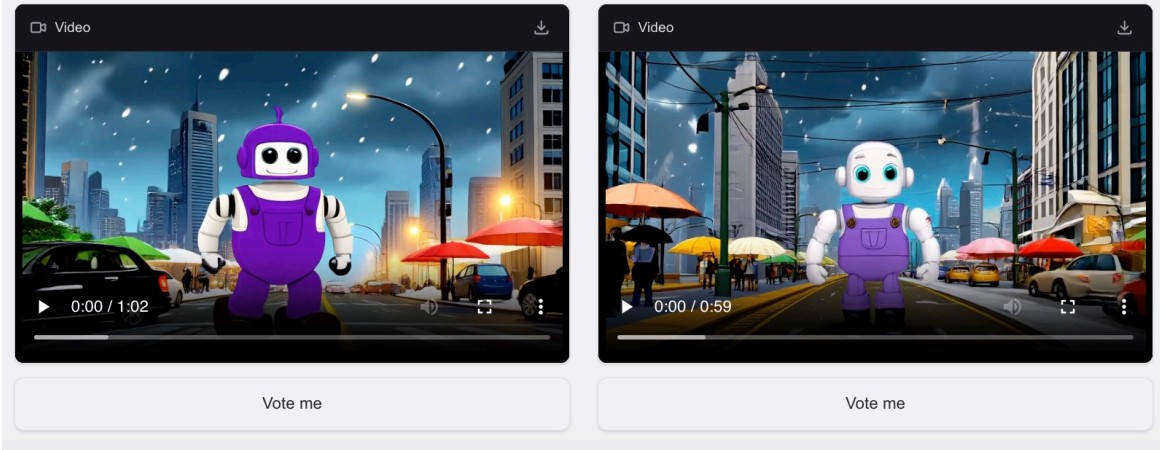

*Figure 11.* **Blind video comparison user study interface.**

