# OpenReview forum: "Context Forcing: Consistent Autoregressive Video Generation with Long Context"
_ICML.cc/2026/Conference — ICML 2026 regular_

### Official Review · Reviewer_3YRp · 2026-03-08

**Soundness:** 2
**Presentation:** 2
**Significance:** 2
**Originality:** 2
**Overall Recommendation:** 2
**Confidence:** 5

**Summary:**

Existing autoregressive video generation models typically employ a "short teacher" to supervise "long students." While student can be supervised the quality of long video generation via rollouts, the inherent student-teacher mismatch leads to the Forgetting-Drifting Dilemma. To address this, the authors propose training a long-context teacher model to supervise student training, thereby enabling the student model to generate more consistent and high-quality videos. Additionally, the authors design a memory mechanism for dynamic context storage. Extensive results demonstrate the effectiveness of the proposed method.

**Compliance With Llm Reviewing Policy:**

Affirmed.

**Final Justification:**

Thank you for the authors’ efforts in the rebuttal. The response addressed part of my concerns, but I still find the improvement brought by the proposed long-video teacher training to be not very noticeable.

My concern is mainly twofold. First, although the trained long-video teacher uses bidirectional modeling within each chunk, its performance on standard VBench metrics is still not as strong as existing models such as Wan 2.1-T2V-1.3B. Second, under the supervision of this long-video teacher, the gain to the student model also appears limited. The learned video dynamics are still quite different from those of standard strong long-video models. More specifically, the generated videos often still give the impression of largely static scenes with repetitive and somewhat meaningless looping motion, rather than convincing long-range temporal dynamics. In this sense, higher consistency does not necessarily mean better quality, especially when the result is close to a nearly static shot. Therefore, I am inclined to maintain my current score.

**Key Questions For Authors:**

I hope the authors can provide more details on the training process of the autoregressive teacher model, its empirical performance, and the motivation-related concerns I raised. I also hope the authors can explain why the dynamics in the final visualizations appear limited, as well as the advantages of the new positional encoding design.

**Limitations:**

The authors include the potential impact statement, but do not explicitly discuss the limitations of the work. I encourage the authors to include a discussion of these limitations.

**Strengths And Weaknesses:**

Strengths:
1. This work bridges the gap in prior methods when using DMD to train long student and teacher models for matching in autoregressive video generation.
2. The model achieves better results on the consistency metric for long-video generation.

weaknesses:
1. Motivation: The fundamental reason why previous works, such as the original Self-Forcing and the CausVid series, utilized asymmetric distillation (i.e., student-long, teacher-short) is based on a specific premise: autoregressive long video training is difficult, whether using Teacher Forcing or Diffusion Forcing. For the DMD process, where distribution consistency is the primary goal, a bidirectional short-clip DiT model can serve as an effective teacher to ensure that short clips from students conform to the data distribution. If long-context autoregressive training were already sufficiently effective, we could directly apply DMD distillation to get a few-steps student models. Therefore, the "gap" this paper seeks to bridge is not a previously undiscovered problem, but rather a known limitation imposed by the capabilities of autoregressive teacher models.

2. Autoregressive Teacher: Following the logic of the first point, since this paper utilizes an autoregressive long video model as the teacher, it is necessary to prove that this teacher model is sufficiently high-performing. However, the paper only briefly introduces the training of the autoregressive teacher model, specifically using a Teacher Forcing approach (albeit with added e_drift perturbations to simulate cumulative historical errors). The term "past model residuals" is also not clearly explained, making it difficult to understand why this teacher model should be considered "good enough." The authors need to provide experimental evidence of the teacher's quality, ideally demonstrating that it outperforms or is comparable to the original bidirectional video DiT models.


3. Linked to the previous point, if the teacher model P_T is not sufficiently trained, the substitution of P_data with P_T in Equations 2 through 5 becomes mathematically tenuous. In the original Self-Forcing framework, the assumption that P_data≈P_T holds because the bidirectional video DiT model can fit the distribution with acceptable error margins.

4. Context Management System: The design of the Context Management System lacks novelty. The implementation of "attention sink" and "fast memory" appears similar to the mechanisms in LongLive. Regarding the "Slow Memory," the historical keyframe selection mechanism has been applied in previous works. For instance, similar selection strategies can be found in: Li, Zhiyuan, et al. "PersonaLive! Expressive Portrait Image Animation for Live Streaming." arXiv preprint arXiv:2512.11253 (2025).

5. Bounded Positional Encoding: The design intuition behind the Bounded Positional Encoding is not clearly articulated. It is difficult to understand why this specific encoding works and what superior characteristics it possesses compared to standard positional indexing.

6. Visualizations of Long-Video Dynamics: Based on the visual results, despite the long teacher model being trained on long-context data, the students supervised by this teacher exhibit similar motion dynamics with LongLive (which uses a short teacher). This does not seem to be a reasonable phenomenon, since long real-world videos would normally be expected to exhibit richer and more coherent dynamics. For example, Cai, Shengqu, et al. "Mode Seeking meets Mean Seeking for Fast Long Video Generation." arXiv preprint arXiv:2602.24289 (2026).

---

> ### Author Rebuttal · Authors · 2026-03-31
>
> Dear reviewer `3YRp`,
>
> We appreciate the reviewer comments. More results are on [anonymous link](https://anonymized1111.github.io/anonymized_ICML2026_rebuttal/).
>
> > **W1 Motivation**
>
> We agree with the reviewer and showed in Fig. 2 that Self-Forcing/CausVid train on short clips(5s), operates in a "Short Bi-dir teacher supervises Short AR student" setting without a supervision length gap, whereas LongLive/Self-Forcing++ rely on a problematic "Short Bi-dir teacher(5s) supervises Long AR student(60s)" setup due to the absence of long teacher.
>
> The motivation of this work is we believe the "short teacher-long student" gap is an urgent problem to address. We aim to raise the community's attention to the importance of a long teacher, as this gap fundamentally bottlenecks long video generation. The student models $P(x_{\text{current}} \mid x_{\text{past}})$, while a memoryless teacher, blind to the past, can only approximate $P(x_{\text{current}})$. This mismatch forces the student to ignore long-range dependencies, leading to suboptimal optimization.
>
> We agree that, "If long-context training were sufficiently effective... could directly apply DMD distillation." We also believe this is true, but to the best of our knowledge, no work systematically confirms whether a long-context teacher indeed provides better supervision than a short teacher, or, if we already have a long teacher, how to effectively distill it. These are all nontrivial questions that require scientific study. Our work provides unique insight by systematically studying those questions. Specifically,
> - Our work directly shows the benefits of using a long teacher: our model trained with a 1.3B long-context teacher matches or surpasses baselines all trained with a 14B short teacher.
> - Most importantly, even if a long teacher exists, how to distill it is still nontrivial. The core contribution of our work, the Context DMD method, is among the first to directly address this.
>
> To summarize our motivation: we present a framework-level study that highlights the importance of addressing the "short teacher–long student" gap, which we believe is currently underexplored in the literature. We will revise the introduction to better reflect the reviewer’s suggestions and clarify this positioning.
>
> > **W2 & W3 Teacher Training Details and Quality**
>
> We first explain how to train a strong teacher, and then demonstrate its effectiveness.
>
> **Training Details:**  For each ground-truth video longer than 10 seconds, we prefill a 21-frame context memory system (3 sink, 12 slow-memory, and 6 fast-memory), aligning with the student's memory system. The subsequent 21 latent frames in the video then serve as the prediction target. To better approximate $P_T \approx P_{data}$ under exposure bias, we apply Error-Recycling Fine-Tuning, where "past model residuals" $e_{\mathrm{drift}}$ are defined as the teacher's own latent prediction errors rather than hand-crafted perturbations. Concretely, at timestep $t$, we obtain the predicted latent $X_{vid}^{pred}$ via one-step ODE integration and define
>
> $e_{drift} = X_{vid}^{pred} - X_{vid}^{*}$,
>
> where $X_{vid}^{*}$ is the clean target latent. These residual tensors are computed online, stored in a timestep-indexed shared buffer, and uniformly resampled and probabilistically injected into the context memory during teacher training. This exposes the teacher to its own historical drift and trains it to recover from it. Please refer to **Fig. 1** on the website for the pipeline.
>
> **Performance:** We have added experiments as suggested, which show that our context teacher performs comparably to the original bidirectional DiT models:
> **Table1** shows quantitative results and **Videos 2** shows qualitative results on website. In our submission, Fig. 6 demonstrates strong context continuation of the teacher, Fig. 7 highlights the importance of Error-Recycling Fine-Tuning, and Tab. 2 reports the teacher's consistency performance.
>
> > **W4 Context Management System**
>
> We thank the reviewer for pointing out this important prior work. We will properly cite (Li, Zhiyuan, et al.) in our final revision.
>
> > **W5 Bounded Positional Encoding**
>
> We apologize for the lack of clarity. In standard RoPE, when inferring sequences longer than the training context, the model encounters OOD unseen position indices, which lead to degraded performance. By bounding the position indices, we ensure they remain within the training distribution regardless of the inference rollout length. This is further validated by Tab. 3 of the paper.
>
> > **W6 Long-Video Dynamics**
>
> We thank the reviewer for highlighting this concurrent work and will cite (Cai et al.) in the revised paper.
>
> We provide dynamic scores at R `zUyE` 's W3 and cases on web.
>
> Notably, our teacher and student are both 1.3B, while LongLive and Cai et al. rely on 14B teachers. (Cai et al. demos use 14B students per Arxiv Sec. 4.1). We expect similar or stronger dynamics when scaling our models to 14B.

---

> > ### Author Rebuttal · Reviewer_3YRp · 2026-04-03
> >
> > Thank you to the authors for the thoughtful responses. I have carefully checked each of them. While some of my concerns have been partially addressed, I still reserve my judgment on several core issues.
> >
> >
> >
> > **W1.** The authors’ original motivation was somewhat confusing, and I appreciate the clarification. As the authors explained, the main focus of this paper is more about “if we already have a long teacher, how to effectively distill it.” However, there are still two key issues that I think have not been adequately resolved.
> >
> >
> >
> > First, the difficulty of training an AR long-video DiT to achieve performance comparable to a bidirectional model is itself a fundamental issue (**W3**). As the authors noted, and as shown in Table 1, the AR teacher performs rather poorly. Compared with Wan 1.3B, the gap is nearly 1.2 points. Therefore, I do not think it is appropriate to describe the performance as comparable.
> >
> >
> >
> > Second, the authors claim that first training a long-video teacher model to supervise the student leads to better results. However, judging from the generated long-video results, the motion dynamics still differ significantly from those in real-world long videos (**W6**). In practice, the results remain quite similar to those produced by short-teacher training methods, such as LongLive. This makes the claimed advantage less convincing. If the proposed method is truly effective, one would expect more natural and realistic long-video motion patterns in the outputs.
> >
> >
> >
> > Finally, regarding **W4**, the authors still do not sufficiently discuss the distinction between their context management system and the methods I mentioned earlier.

---

> > > ### Author Response · Authors · 2026-04-06
> > >
> > > Dear reviewer `3YRp`,
> > >
> > > Thanks for your thoughtful response.
> > >
> > > > Performance of long context teacher
> > >
> > > We agree with the reviewer that training a long AR model is hard. To clarify, our long context teacher is a bidirectional video continuation model. Therefore, in the submission and rebuttal, we refer to it as long-context teacher rather than long AR teacher. It takes a long context video as input and then **predict 5s video chunk in a single shot** use **bidirectional attention** exactly the same as standard DiT and it does not generate the 5s chunk in autoregressive way. So this model is much easilier to train than purely AR model.
> > >
> > > We further show the effectiveness of this teacher by 4 experiments:
> > >
> > > 1. On VBench 5s setting, compared with Wan 1.3B, our teacher model outperforms it by 0.67 and 0.72 points on background and subject consistency. This aligns with expectation that models trained with long video data exhibit stronger consistency. It performs slightly worse on semantic metrics, which is expected given our fine-tuning data is significantly smaller in scale than Wan 1.3B’s pretraining data.
> > >
> > > 2. Following Cai et al., we use Gemini-3-Pro with the same instructions and metric (VLM consistency) from their paper to evaluate our model on 100 text prompts.
> > >
> > > In 30s setting. Our teacher outperforms existing long-video methods in VLM consistency:
> > >
> > > | Method | VLM consistency↑ |
> > > | :--- | :---: |
> > > | Self Forcing(1.3B) (14B teacher) | 30.39 |
> > > | LongLive(1.3B) (14B teacher) | 54.22 |
> > > | Infinity-RoPE(1.3B) (14B teacher) | 53.83 |
> > > | **Ours Teacher(1.3B)** | **64.20** |
> > > | **Ours Student(1.3B) (1.3B teacher)** | 62.53 |
> > >
> > > In 5s setting. our teacher outperforms Wan2.1-T2V-1.3B.
> > >
> > > | Method | VLM consistency↑ |
> > > | :--- | :---: |
> > > | Wan2.1-T2V-1.3B | 66.54 |
> > > | **Ours Teacher(1.3B)** | **67.53** |
> > >
> > > 3. We conduct additional human eval on 5s video generation, with same setting described in reply to Rfia9(User study's UI shown on web). We have 7 volunteers. Our teacher outperforms Wan2.1-T2V-1.3B.
> > >
> > > | Comparison | Ours Win (%) ↑ | Ours Lose (%) ↓ |
> > > | :--- | :---: | :---: |
> > > | **Ours Teacher** vs. Wan2.1-T2V-1.3B | **61.43** | **38.57** |
> > >
> > >
> > > 4. We provide visualizations on website. The long-context teacher acts as an effective video continuation model, generating natural motion conditioned on real video inputs.
> > >
> > >
> > > Therefore, based on these 4 experiments, we show that our 1.3B long-context teacher achieves performance comparable to Wan2.1-T2V-1.3B.
> > >
> > >
> > > > Our results motion dynamics
> > >
> > > We thank the reviewer for raising this question.  Our claim in the submission is that Context Forcing preserves superior consistency over long durations. We do not claim that Context Forcing improves motion dynamics. Our experiments in the submission are designed to directly support this claim. In particular, the results consistently show improved long-term consistency compared to LongLive.
> > >
> > > Regarding motion dynamics, we believe this aspect is largely constrained by the model’s capacity (size) and pretraining data, and therefore cannot be significantly improved by a distillation method alone.
> > > In our 1.3B setting, motion quality is bounded by the base model, while Context Forcing primarily improves consistency. On the other hand, long video data does not necessarily introduce high dynamics; long videos can be slow motion, such as the slow-motion ballet demo in Cai et al. Therefore, we do not expect the model to learn larger motion using a long-context teacher.
> > >
> > > The highlighted concurrent work (Cai et al.) demos, which the reviewer thinks have better motion, are generated by a 14B model fine-tuned on 100K high-quality video data (according to Section 4.1), while ours is a 1.3B student achieved by a distilled method from a 1.3B teacher on synthesized student rollouts. We therefore believe these results are not directly comparable.
> > >
> > >
> > > > Comparing with concurrent work
> > >
> > >
> > > We thank the reviewer for raising this question. The only similarity between Li et al.’s historical keyframe mechanism and our context management system lies in the high-level idea of similarity-based keyframe selection, while the implementations differ substantially. To enable low-latency selection, we compare similarity directly on K in the KV cache, whereas they rely on a motion extractor to derive motion features. In addition, their distillation phase does not incorporate the memory(keyframes) mechanism into the teacher model. Furthermore, their method does not include bounded RoPE for the memory system or a slow-fast memory design (i.e., it does not separate historical keyframes into slow and fast memory). Architecturally and functionally, their approach is based on a U-Net rather than a DiT, does not use DMD, and targets streaming animation, which differs fundamentally from our setting.

---

### Official Review · Reviewer_zUyE · 2026-03-10

**Soundness:** 3
**Presentation:** 3
**Significance:** 3
**Originality:** 3
**Overall Recommendation:** 4
**Confidence:** 4

**Summary:**

This paper presents a novel framework for achieving long-term consistency in autoregressive video generation. It introduces the Context Forcing framework to resolve the inherent student-teacher mismatch in existing methods, and designs a novel Slow-Fast Memory architecture that enables the robust training of models with strong long-term temporal consistency. Extensive experiments demonstrate that the proposed model delivers outstanding empirical performance on long video generation tasks.

**Compliance With Llm Reviewing Policy:**

Affirmed.

**Final Justification:**

The authors’ rebuttal and additional results addressed my concerns. I therefore maintain my overall positive recommendation of Weak accept.

**Key Questions For Authors:**

Please refer to weaknesses

**Limitations:**

yes

**Strengths And Weaknesses:**

### Strengths：
- The Slow-Fast Memory architecture is a novel and highly practical contribution; it enables extreme long-context video generation (20+ seconds) while avoiding prohibitive computational costs, which is a standout merit of this work.

- The paper is well-written with a clear logical flow, making the technical content easy to follow and comprehend.

- The proposed Context Forcing framework and Slow-Fast Memory system are built on standard autoregressive diffusion architectures (e.g., DiT) and KV caching—components that are widely used in modern generative AI. This makes the method easy to implement and adapt to other text-to-video or video continuation models, with clear potential for real-world applications.

### Weaknesses：
- The empirical evaluation focuses on general video generation but does not test Context Forcing on highly dynamic or complex scenes (e.g., fast camera motion, multiple moving subjects, scene transitions). It is unclear how the method performs in these challenging settings, where the demand for long-term context and semantic retention is even higher. A qualitative/quantitative evaluation on such scenarios would highlight the model’s robustness in real-world use cases.
- The paper states that the Slow-Fast Memory system makes long-context generation computationally feasible but provides only limited details on inference time and computational overhead. While the paper reports throughput for short video generation, it does not quantify how throughput scales with longer context lengths (e.g., 20s vs. 60s video). This information is critical for real-time applications.
- In the visualization results of the paper, the motion variations appear to be quite minimal. Additionally, there seems to be no quantitative evaluation of motion-related metrics in the experimental results. Could the authors provide comparative results for motion metrics to further validate the model's performance in capturing dynamic temporal changes?
- Minor: There is a typo in Line 78: "FIgure5" should be corrected to "Figure 2".

---

> ### Author Rebuttal · Authors · 2026-03-31
>
> Dear Reviewer `zUyE`,
>
> We thank the reveiwer for the encouraging feedback and recognizing the novelty and practical potential of our Context Forcing framework and Slow-Fast Memory architecture. We put more results on [anonymous link](https://anonymized1111.github.io/anonymized_ICML2026_rebuttal/).
>
> We address the raised issues below:
>
> > **W1: Evaluation on highly dynamic or complex scenes.**
>
> As suggested, we have conducted additional quantitative and qualitative experiments specifically targeting highly dynamic scenes (e.g., fast camera motion, multiple moving subjects, and scene transitions).
> * **Quantitative:** We evaluated these scenarios using VBench. The motion-related metric scores are provided in the table under our response to **W3**.
> * **Qualitative:** We have compiled over high-dynamic video cases and uploaded them to our [anonymous link](https://anonymized1111.github.io/anonymized_ICML2026_rebuttal/) **Videos 1** part for your review.
>
> > **W2: Throughput and computational overhead scaling with longer context.**
>
> As suggested, we have supplemented our analysis to clarify how inference throughput scales with the effective context length rather than the raw video length. In our Slow-Fast Memory design, the Slow Memory dynamically samples key frames. Therefore, the computational overhead is bounded by the total size of the retained KV cache (Sink + Slow + Fast), not the total number of frames in the video. As shown in the table below, this selective retention decouples the memory footprint from the temporal length, maintaining competitive throughput even for extended videos (20s+). We will include this detailed breakdown in the final manuscript.
>
> | Total Context Length | Total KV Length| Sink Size | Slow Memory Size | Fast Memory Size | Throughput (FPS) ↑ |
> | :--- | :---: | :---: | :---: | :---: | :---: |
> | 27.0s+ | 24 | 3 | 15 | 6 | 15.9 |
> | 22.5s+ | 21 | 3 | 12 | 6 | 17.0 |
> | 18.0s+ | 18 | 3 | 9 | 6 | 18.2 |
> | 13.5s+ | 15 | 3 | 6 | 6 | 19.6 |
>
> > **W3: Quantitative evaluation of motion-related metrics.**
>
> While motion metrics were included in our ablation study (Table 3) in the submission, we further address this concern by conducting an additional comprehensive baseline comparison on VBench, explicitly evaluating general motion variations and dynamic temporal changes.
>
> As shown in the table below, our method achieves the best dynamic performance results. Notably, our approach only uses a lightweight 1.3B teacher model, while all baseline methods rely on 14B teachers, this further underscoring the advantage of our Context Forcing method. Additional visualizations of these dynamic capabilities can be found in the supplementary video and on our [anonymous link](https://anonymized1111.github.io/anonymized_ICML2026_rebuttal/).
>
> | Method | Dynamic Degree on 5s↑ | Dynamic Degree on 60s↑ |
> | :--- | :---: | :---: |
> | LongLive (14B teacher) | 0.54 | 0.56 |
> | Infinity-RoPE (14B teacher) | 0.48 | 0.52 |
> | Rolling Forcing (14B teacher) | 0.40 | 0.36 |
> | **Ours (1.3B teacher)** | **0.56** | **0.58** |
>
> > **W4: Typo in Line 78.**
>
> We have corrected "Figure5" to "Figure 2" in the revised manuscript.

---

> > ### Author Rebuttal · Reviewer_zUyE · 2026-04-03
> >
> > Thank you for your detailed rebuttal and the additional results provided. They have addressed my concerns, and I maintain my positive recommendation of weak accept.

---

> > > ### Author Response · Authors · 2026-04-06
> > >
> > > Dear Reviewer `zUyE`,
> > >
> > > Thanks for your response. We appreciate your comments and acknowledgment. We hope our response has addressed all of your concerns. We would be grateful if you would consider raising your overall recommendation in light of our clarifications.
> > >
> > > Thank you,

---

### Official Review · Reviewer_fia9 · 2026-03-13

**Soundness:** 3
**Presentation:** 4
**Significance:** 4
**Originality:** 3
**Overall Recommendation:** 4
**Confidence:** 3

**Summary:**

This paper proposes a new approach called ContextForcing for auto-regressive long video generation. Instead of distilling long student from short teacher which causes potential teacher-student mismatch, the proposed paper extend teacher to long teacher (with context modeling) and distills long student from long teacher. The paper further introduces a Slow-Fast Memory management system to encode long context. The resulting model can generate high quality results (>20s long) outperforming recent methods.

**Compliance With Llm Reviewing Policy:**

Affirmed.

**Final Justification:**

The rebuttal addressed my concerns pretty well, so I'd like to keep my positive score.

**Key Questions For Authors:**

No additional questions.

**Limitations:**

Yes.

**Strengths And Weaknesses:**

Strengths:
1. The paper is very well-written, and easy to follow.
2. The main ideas  (training long context teacher with video continuation task and slow-fast memory architecture) seem technically sound, interesting and novel.
3. Extensive experiments and ablation studies are conducted to demonstrate effectiveness and advantage of the proposed method.

Weaknesses:
1. From Tab.2, the quantitative results seem the scores of "Total" and "Quality" are worse than LongLive while "Semanitc", "Bg Consistency", "Sub consistency" are better. Overall, it seems not obviously better.
2. Listing a visual failure cases of LongLive is not convincing. It would be great to have user study on visual evaluation between proposed method and LongLive.

---

> ### Author Rebuttal · Authors · 2026-03-31
>
> Dear Reviewer `fia9`,
>
> Thank you for your constructive feedback and for your positive evaluation of our paper. We appreciate your recognition of the clarity of our writing, the novelty of our Context Forcing approach, the Slow-Fast memory design, and the strength of our experimental results. To further support our claims, we have included additional results at [anonymous link](https://anonymized1111.github.io/anonymized_ICML2026_rebuttal/).
>
> Below, we address the rasied concerns in detail and clarify the key points raised.
>
> > **W1: "Total" and "Quality" results are worse than LongLive**
>
> We agree that our "Total" and "Quality" scores are comparable to, rather than strictly outperforming, LongLive [1]. To further address the reviewer’s concern, we clarify the trade-offs involved and highlight our method’s clear advantages in consistency and effectiveness:
>
> * **Global Consistency over Local Quality:** While LongLive achieves marginally higher frame-level quality scores, it suffers from severe global degradation, notably the "flashback" issue (**as illustrated in our paper Fig. 8 and discussed in Line 372**). This means that while individual short clips might look high-quality, the overall long video completely loses temporal coherence. This exact limitation of LongLive was also independently reported in a recent concurrent work, LoL [2]. Our method, as reflected by our superior "Semantic," "Bg Consistency," and "Sub consistency" scores, resolves this to maintain global coherence across the entire generated video.
> * **Model Scale:** It is also crucial to note the significant difference in model capacity. LongLive relies on a 14B parameter teacher model, whereas our Context Forcing model achieves these highly competitive and even better results using only a 1.3B parameter teacher model. We believe achieving superior long-context consistency at a fraction of the parameter count futher demonstrates the core effectiveness of our approach.
>
> We provide visualization and comparision with LongLive, please refer to **Videos 3** on our [anonymous link](https://anonymized1111.github.io/anonymized_ICML2026_rebuttal/).
>
> *[1] Yang, Shuai, et al. "Longlive: Real-time interactive long video generation." arXiv preprint arXiv:2509.22622 (2025).*
>
> *[2] Cui, Justin, et al. "LoL: Longer than Longer, Scaling Video Generation to Hour." arXiv preprint arXiv:2601.16914 (2026).*
>
> > **W2: Additional User study.**
>
> As suggested, we conduct an additional user study and provide comprehensive visual evaluations to further substantiate our claims.
>
> **Exp: User Study on Visual Quality and Consistency**
> During the rebuttal period, we designed and conducted a blind A/B testing user study via a web interface with 15 participants. We randomly selected 20 text prompts and generated corresponding long videos using LongLive, Infinity-RoPE [3], and our Context Forcing method. Participants were shown paired videos (Ours vs. Baseline) in a randomized order. To guide their evaluation, participants were asked to comprehensively consider (1) overall visual quality, (2) long-term temporal consistency, and (3) semantic alignment, and then select their single most preferred result.
>
> The overall preference (win rate) is summarized in the table below, demonstrating a clear user preference for our method due to its stable long-term generation:
>
> | Comparison | Ours Win (%) ↑ | Ours Lose (%) ↓ |
> | :--- | :---: | :---: |
> | **Ours** vs. LongLive | 69.3 | 30.7 |
> | **Ours** vs. Infinity-RoPE | 77.3 | 22.7 |
>
> We also provide our user study UI and final result, please refer to **Figure 2&3** on our [anonymous link](https://anonymized1111.github.io/anonymized_ICML2026_rebuttal/).
>
> We will include these user study details, along with the detailed web interface setup, in the appendix of the final revision. We hope these additional results and clarifications address your concerns!
>
> *[3] Yesiltepe, Hidir, et al. "Infinity-rope: Action-controllable infinite video generation emerges from autoregressive self-rollout." arXiv preprint arXiv:2511.20649 (2025).*

---

> > ### Author Rebuttal · Reviewer_fia9 · 2026-04-03
> >
> > The rebuttal has addressed most of my concerns. I will keep my rating.

---

> > > ### Author Response · Authors · 2026-04-06
> > >
> > > Dear Reviewer `fia9`,
> > >
> > > Thanks for your response. We appreciate your comments and acknowledgment. We hope our response has addressed all of your concerns. We would be grateful if you would consider raising your overall recommendation in light of our clarifications.
> > >
> > > Thank you,

---

### Decision · Program_Chairs · 2026-04-30

**Decision:**

Accept (regular)

**Comment:**

This paper proposes Context Forcing, a framework of training a long-context teacher to supervise autoregressive video generation students, addressing the student-teacher mismatch in prior methods where short (5s) teachers supervise long (60s) students. A Slow-Fast Memory architecture manages the growing context efficiently. Scores are 4/4/2.

**Strengths**:
  - The framework addresses a clearly motivated and underexplored problem: prior streaming methods cap the student's effective context by using memoryless short-clip teachers.
  - The Slow-Fast Memory design is practical and well-engineered, decoupling memory cost from video length.
  - Strong consistency results: the method outperforms LongLive and Infinity-RoPE on background/subject consistency and VLM consistency metrics, and is preferred 69.3% and 77.3% of the time in human evaluation — notably using only a 1.3B teacher while all baselines use 14B.

**Reviewer 3YRp's concerns and my assessment**:

Reviewer 3YRp raised two main issues. First, the long-context teacher underperforms Wan 1.3B on VBench by ~1.2 points. The authors addressed this during discussion with four additional experiments: the teacher outperforms Wan 1.3B on consistency-specific VBench metrics, achieves higher VLM consistency scores (Gemini-evaluated), and is preferred over Wan 1.3B 61.4% of the time in human evaluation. The VBench gap appears driven by semantic metrics, which is expected given the smaller fine-tuning data.

Second, Reviewer 3YRp noted that motion dynamics remain similar to LongLive. The authors stated that their claim is about long-range consistency, not motion dynamics. Motion quality is bounded by model capacity and pretraining data, not the distillation method. The comparison to Cai et al. (a 14B model on 100K curated videos, released after submission) is not directly applicable. I agree improving consistency without degrading dynamics at 1.3B scale is itself a meaningful contribution.

The context management system shares the high-level idea of similarity-based keyframe selection with concurrent work (Li et al.), but differs substantially in implementation (KV-cache-based selection, slow-fast separation, bounded RoPE, DiT+DMD setting). Li et al. appeared publicly on Dec 12, 2025, within the concurrent work window under ICML 2026 policy.

In summary, the paper makes a clear contribution to long-context video generation with a well-motivated framework, practical memory design, and strong consistency results at a fraction of baseline model size. Reviewer 3YRp's concerns about motion dynamics, while valid observations, do not contradict the paper's actual claims. I recommend acceptance, with the authors incorporating the additional teacher evaluations and human study into the camera-ready.